# A Machine Learning Application to Predict Early Lung Involvement in Scleroderma: A Feasibility Evaluation

**DOI:** 10.3390/diagnostics11101880

**Published:** 2021-10-12

**Authors:** Giuseppe Murdaca, Simone Caprioli, Alessandro Tonacci, Lucia Billeci, Monica Greco, Simone Negrini, Giuseppe Cittadini, Patrizia Zentilin, Elvira Ventura Spagnolo, Sebastiano Gangemi

**Affiliations:** 1Department of Internal Medicine, Scleroderma Unit, Clinical Immunology Unit, University of Genoa, 16143 Genoa, Italy; giuseppe.murdaca@unige.it (G.M.); monica.greco@unige.it (M.G.); simone.negrini@unige.it (S.N.); 2Radiology Unit, IRCCS Policlinico San Martino, 16132 Genoa, Italy; simone.caprioli@hsanmartino.it (S.C.); giuseppe.cittadini@hsanmartino.it (G.C.); 3Clinical Physiology Institute, National Research Council of Italy (IFC-CNR), 56124 Pisa, Italy; lucia.billeci@ifc.cnr.it; 4Department of Internal Medicine, Gastroenterology Unit, University of Genoa, 16143 Genoa, Italy; patrizia.zentilin@unige.it; 5Section of Legal Medicine, Department of Health Promotion Sciences, Maternal and Infant Care, Internal Medicine and Medical Specialties (PROMISE), University of Palermo, Via del Vespro, 129, 90127 Palermo, Italy; elvira.ventura@unipa.it; 6Department of Clinical and Experimental Medicine, School and Operative Unit of Allergy and Clinical Immunology, University of Messina, 98122 Messina, Italy; gangemis@unime.it

**Keywords:** artificial intelligence, esophageal dilatation, HRCT chest, machine learning, systemic sclerosis

## Abstract

Introduction: Systemic sclerosis (SSc) is a systemic immune-mediated disease, featuring fibrosis of the skin and organs, and has the greatest mortality among rheumatic diseases. The nervous system involvement has recently been demonstrated, although actual lung involvement is considered the leading cause of death in SSc and, therefore, should be diagnosed early. Pulmonary function tests are not sensitive enough to be used for screening purposes, thus they should be flanked by other clinical examinations; however, this would lead to a risk of overtesting, with considerable costs for the health system and an unnecessary burden for the patients. To this extent, Machine Learning (ML) algorithms could represent a useful add-on to the current clinical practice for diagnostic purposes and could help retrieve the most useful exams to be carried out for diagnostic purposes. Method: Here, we retrospectively collected high resolution computed tomography, pulmonary function tests, esophageal pH impedance tests, esophageal manometry and reflux disease questionnaires of 38 patients with SSc, applying, with R, different supervised ML algorithms, including lasso, ridge, elastic net, classification and regression trees (CART) and random forest to estimate the most important predictors for pulmonary involvement from such data. Results: In terms of performance, the random forest algorithm outperformed the other classifiers, with an estimated root-mean-square error (RMSE) of 0.810. However, this algorithm was seen to be computationally intensive, leaving room for the usefulness of other classifiers when a shorter response time is needed. Conclusions: Despite the notably small sample size, that could have prevented obtaining fully reliable data, the powerful tools available for ML can be useful for predicting early lung involvement in SSc patients. The use of predictors coming from spirometry and pH impedentiometry together might perform optimally for predicting early lung involvement in SSc.

## 1. Introduction

Systemic sclerosis (SSc) is a systemic immune-mediated disease, characterized by fibrosis of skin and organs, featuring one of the greatest mortality ratios among rheumatic diseases [1]. Recent studies have demonstrated the involvement of peripheral and autonomic nervous systems in SSc, in turn featuring white matter lesions in patients, even when asymptomatic, so that the involvement of smaller branches and perforating arteries was hypothesized [2]. However, lung involvement is deemed to be the leading cause of death in SSc, overall. Indeed, as with the skin, lungs are also affected by fibrosis and interstitial lung disease (ILD), with non-specific interstitial pneumonia (NSIP) on high resolution computed tomography (HRCT) being the most common manifestation, without the need to undergo lung biopsy in such cases [3]. As a whole, ILD—often recognized as being the first cause of SSc-related death [4]—consists of a group of diseases that affects lung interstitium, involving alveolar epithelium, pulmonary capillary endothelium, basement membrane and perivascular and perilymphatic tissues. A complex interplay between innate and adaptive immunity, endothelial dysfunction, small vessel abnormalities and inflammation plays a fundamental role in the onset and maintenance of fibrosis. Within this frame, the damage to the alveolar and vascular endothelial epithelial cells, and the consequent profibrotic stimuli that induce the differentiation of pulmonary fibroblasts to the myofibroblast phenotype, represents the trigger of the pathogenesis of SSc-ILD [5,6,7,8].

Notably, from the clinical point of view, one of the most important challenges in the management of this pathology is represented by the risk stratifications of complications and, ultimately, death; as a matter of fact, although no disease-modifying drugs have been found in SSc, early screening and management of patients improves survival [3]. However, elective tests for large-scale screening of SSc-ILD, represented by pulmonary function tests, are not sensitive enough to be used for screening purposes [9], in turn raising a significant problem in the definition of affordable and reliable screening methods for this specific diagnostic question.

To this extent, technological advances can help the clinicians screening the most functional diagnostic methods to be picked up, supporting diagnosis and, in some instances, clinical decision.

In this framework, Artificial intelligence (AI) will probably play the starring role in the future of medicine. Indeed, increasing growth in the interest towards AI is seen in (bio)-medical research, driving valuable outputs in the framework of the “p4 medicine” [10]. Notably, Machine learning (ML) is one of the most exciting fields of research in AI and computer science and it is increasingly used in medical research. The main characteristic of ML is that the computer is given the ability to learn without being explicitly programmed to do it. ML algorithms can use a large amount of data and extract meaningful results from them. As such, ML can be used in several applications: for classifying patients, predicting future outcomes and even individualize patients’ treatment [11,12,13].

Specifically, within the field of immune-mediated diseases and, more in depth, considering the SSc as the reference model, the application of AI for supporting the diagnosis of SSc can confirm that the possibility exists for the early prediction of ILD, anticipating functional signs shown by the spirometry and pH-impedentiometry. This would eventually enlarge the indication of HRCT foreseen in the International guidelines [14,15].

Considering the existing literature in the field, ML was employed in identifying SSc in electronic records. Notably, Jamian and colleagues [16], using rule-based and ML techniques, applying classification and regression tree (CART) and random forest (RF) for algorithm development, were able to identify SSc from a large Electronic Health Record with an overall accuracy of up to 90%; however, it was lacking portability and generalizability according to the authors. Skin biopsies were otherwise used by Franks and colleagues [17] to classify patients with SSc depending on their intrinsic molecular subsets. The task was performed using supervised classifiers and resulted in a successful result in over 85% of cases in the case of the multinomial elastic net (GLMnet), displaying the highest classification values of all, with the authors trying to cope at best with the limitations derived from having a small sample size with a significant number of features to be included in the model. However, as observed [18], the authors used multiple, small and not always comparable datasets for the test set that, added to the likely overfitting issues normally experienced in the presence of small amounts of data (particularly on low number of patients), would make the results not necessarily reliable and generalizable.

The association between biomarkers and phenotype of the disease using AI was investigated in some works published to date, displaying overall good outcomes, but leaving room for further investigation, especially in terms of the amount of the biomarkers taken into account. More specifically, Huang and colleagues [19] applied conditional random forests (CRF) coupled with gene set enrichment analysis (GSEA) to identify variables from flow cytometry, which were effective in classifying ILD patients and stochastic simulation to train and validate ILD screening tools.

In light of the growing demand from the clinical world, and considering the current diagnostic gaps experienced in the specific applications of the SSc, in the present research we used ML to predict early lung involvement in SSc from both clinical and instrumental data—for the first time using together both these categories—evaluating this promising approach for early diagnosis in the framework of personalized medicine.

## 2. Methods

### 2.1. Patients

We retrospectively evaluated 38 patients at the Department of Internal Medicine, University of Genoa, and at the Radiology Unit, IRCCS Policlinico San Martino, Genoa. All patients signed written informed consent in compliance with the regulations of the host institution and were treated in accordance with the Declaration of Helsinki.

We collected clinical data from esophageal evaluation (esophagogastroduodenoscopy, esophageal pH impedance test, esophageal manometry and reflux disease questionnaire) and pulmonary evaluation (pulmonary function tests) once for each patient. Of note, the reflux disease questionnaire (RDQ) allowed the evaluation of reflux esophagitis, collecting information about the frequency and severity of upper gastrointestinal symptoms (heartburn, regurgitation and non-cardiogenic chest pain).

Radiological data concerning esophageal diameter and severity of lung involvement were also collected on HRCT images. The widest esophageal diameter (dmax) and the widest esophageal area (area max) were calculated by placing a freehand region of interest (ROI) upon the esophageal contours on para-axial images perpendicular to esophageal lumen where the widest esophageal sectional area was visually assessed. Esophageal diameter and area measurements were also conducted at 30 mm (D3), 50 mm (D5) and 70 mm (D7) from lower esophageal sphincter (LES), corresponding to the points on which the impedance pH test is conducted. LES position was conventionally set at the level of the diaphragmatic plane.

Lung severity assessment was evaluated by a radiology resident blinded to the pulmonary function tests. The Warrick score (WS), a semiquantitative score based on pulmonary anomalies on HRCT [20], was used, and represented the (continuous) outcome of the supervised ML techniques trained in the present work. WS is calculated on the appearance of HRCT anomalies and their extent. Severity was scored from 0 to 5 for each pulmonary segment, while extension was scored for every detected anomaly from 1 to 3 (1 representing 1–3 segments involved, 2 representing 4–9 segments involved and 3 representing more than 9 segments involved). The total extension was calculated as the sum of the extension of every single anomaly. The final WS was calculated as the sum of the severity score and the extension score (from 0 to 30).

Pulmonary involvement was defined upon the presence of signs of interstitial lung disease on HRCT with 6 out of 24 patients having interstitial lung disease on HRCT.

The overall dataset employed, with means and standard deviations (SDs), ranges and distribution type of the 23 variables, including the outcome, is shown in Table 1.

### 2.2. Machine Learning

As briefly mentioned above, the Machine Learning part was aimed at identifying how the clinical variables were predictive of the clinical outcome, calculated as the WS defined in the previous paragraph, and suggestive of the severity of the clinical condition.

To setup the ML approach, we programmed the R language and used the open source RStudio (Boston, MA, USA), version 1.3.1093 for Windows, available with the GNU Affero General Public License.

The dataset employed for ML was complete, without occurrence of not-a-number (NaN) items, therefore not requiring any kind of mitigation measure for missing values. Prior to the ML training phase, we identified the dataset outliers, defined as the values outside the physiological ranges for each variable. After outlier removal, resulting missing data were imputed using the multiple imputation by chained equation (MICE) method. As highlighted by Table 1, the vast majority of the variables considered had a non-normal distribution; therefore, and in order to avoid feature selection biases, a data normalization step for all non-normal variables was performed.

Given the relatively low number of subjects involved in the study, and in order to comply with the basic “rule of thumb” [21,22] applied to ML problems, foreseeing at least 10 observations for each variable of the dataset, we decided to add some further observations stemming from the original ones and added random noise to them, a procedure demonstrated to be particularly useful in the presence of small samples [23]. This procedure, applied limitedly to the training set, was performed using the MATLAB (The MathWorks, Inc., Natick, MA, USA)-based function *randn*.

With the restructured dataset, we drafted and trained five different ML models, all of them supervised, which are briefly reported below. All the models were implemented using the R-based *caret* package [24], a powerful package able to manage different ML models using the same basic options, therefore allowing for a fair comparison between them. The choice for these specific models was performed since the dataset was complete of both predictors (22 variables) and clinical outcome (the WS), therefore requiring a “supervised” approach to ML; in addition, among supervised models, we selected some of the simplest, less computationally burdensome, and widely used algorithms that can be implemented using the *caret* package in order to provide a reliable comparison between the different models using the same training strategy, the same seed for splitting data into a training and test set and the same conditions for training the net.

Prior to the model training, the dataset, composed of 23 variables, including the outcome, and 228 observations, was randomly divided into a fully independent training and test set, with a percentage of 90 and 10% of data, respectively. A 10-fold cross validation was employed in order to obtain robust data with respect to eventual overfitting phenomena (Figure 1).

The models were compared in terms of regression capability according to the root mean square error (RMSE) and to the R-squared, calculated over the test set (and also verified over the training set to evaluate the good quality of training algorithms) with respect to the outcome, and for each ML algorithm, the best model selected was not the one with the lowest RMSE or R-squared, but the simplest acceptable one, represented by the simplest model within one standard error from the lowest RMSE. This choice was performed under two main principles: (i) in order to cope with the trade-off between simplicity and accuracy of the model, and (ii) to further avoid overfitting issues that could eventually decrease the generalizability of the model.

#### 2.2.1. LASSO

The least absolute shrinkage and selection operator, mostly known as LASSO, represents a regression analysis method, often used in the ML framework, performing both variable selection and regularization aimed at enhancing the prediction accuracy and interpretability of the resulting model. Introduced in the ML universe by Tibshirani [25], it becomes particularly useful in the presence of datasets with several variables hypothesized as not being useful for the prediction.

#### 2.2.2. RIDGE

Ridge regression is a widely used technique for analyzing multiple regression data severely affected by multicollinearity problems [26]. If multicollinearity occurs, least squares estimates are completely unbiased, although their variances are large, making them far from their true value. By adding a degree of bias to the regression estimates, ridge regression is able to reduce the standard errors. Conversely to LASSO, with whom it shares several common features, RIDGE regression shrinks all the coefficients to a non-zero value.

#### 2.2.3. Elastic Net

Merging characteristics of both LASSO and RIDGE methods, the elastic net seeks to put together the advantages of both techniques making a blend of them [27]. Its main regularization parameter, named α, can be continuously varied between 0 and 1, with the lower limit making the model equal to RIDGE and the upper one to LASSO. A 0.5 value indicates a 50/50 blend between the two regression models.

#### 2.2.4. CART

Classification and regression trees (CART) are useful and commonly used ML models, based on the deconstruction of the overall sample into smaller groups, performed through repeated, binary splits of the patient sample, considering one exploratory variable at a time [28].

Their advantages are manifold, including the ease of adaptation to different data, including cross sectional, longitudinal, survival data, the possibility to use different types of response variables, and the fact they do not need to make any assumptions in terms of the normality of the data distribution. The main limitations of CART models include their sensitivity to data changes and their somewhat limited interpretability.

#### 2.2.5. Random Forest

Random forest (RF) are learning methods particularly useful for classification and regression, operating by building up a series (forest) of decision trees at the training and outputting the class that is the mode of the classes, for classification, or the mean prediction, for regression, of the individual trees [29].

With respect to the classical decision trees, RF represent an improvement in terms of overfitting issues for the training set. They carry on several advantages, including the performance of implicit on-the-run feature selection, the provision of accurate indicators of feature importance, the absence of need for particular data preparation prior to the application of the ML model, the opportunity for them to handle binary, categorical, numerical features without any need for scaling, normalization or standardization. They are also relatively quick to train and versatile, although their interpretability is often cumbersome.

## 3. Results

### Prediction Accuracy

The five ML methods mentioned above were compared in terms of prediction accuracy, using the RMSE as the main evaluation metrics.

The results obtained are displayed in Table 2.

As displayed, the random forest, which is trained by the R-based *caret* package relying on 500 trees, has the best performances, outperforming the other classifiers, three of which (LASSO, RIDGE and elastic net) display very similar error values performing the regression task.

However, the models were also evaluated depending on other parameters, including the time elapsed for the full training of the regressor, the memory used, and the number of variables included within the model (out of the 22 variables available within the dataset inputs).

Table 3 displays this information.

The random forest model when trained achieved the best performances using a relatively low number of variables (*n* = 7), making it useful also as a predictive tool for the clinician.

A plot of the related RMSE, based on the number of variables included in the model, is displayed in Figure 2.

According to the results obtained, the optimal number of predictors for minimizing the regression error is between 7 and 16, with a minimum of the RMSE reached for 11 variables. However, the choice to use the value of 7 for the optimal model aims at managing the trade-off between model complexity and prediction accuracy, without particular risks concerning overfitting and, therefore, poor generalizability to new datasets.

For the knowledge and use of the clinician, the 7 parameters estimated to have the greater classification importance among the 22 inputs of the current dataset include (in order of importance): (i) total lung capacity (TLC); (ii) mean nocturnal basal impedance at 3 cm (MNBI3); (iii) diffusing capacity for carbon monoxide (DLCO); (iv) forced expiratory volume in the first second (FEV1); (v) forced vital capacity (FVC); (vi) mean nocturnal basal impedance at 5 cm (MNBI5); (vii) mean nocturnal basal impedance at 7 cm (MNBI7).

As mentioned above, the *caret* package used for training allows building up the random forest based on 500 trees, but the result obtained suggests an overall stabilization of the RMSE already at around 150 trees, suggesting that even simpler models, with respect to the one used for the present investigation, would enable a similarly precise regression given the inputs of the present dataset. Therefore, although the employment of the *caret* package was aimed at comparing the different classifiers, other R-based packages, allowing the use of simpler classifiers, can be applied in case of computational constraints eventually occurring.

## 4. Discussion

ILD is the leading cause of death in patients with SSc. Lung fibrosis is present in 80% of patients with SSc; among them, 25–30% develops a progressive ILD. ILD generally presents during the first 4–6 years after the onset of scleroderma [3]. Since ILD is a life threatening and early complication and a new therapeutic approach can be used [30], early screening tests are needed. Generally, a baseline HRCT is used to detect ILD in newly diagnosed SSc. The most frequent HRCT pattern of lung involvement in SSc is NSIP, with bilateral ground glass opacities as a dominant feature, along with fibrotic reticular changes and traction bronchiectasis; less frequently, a usual interstitial pneumonia (UIP) pattern is observed [31,32,33]. Some authors also questioned if interstitial lung abnormalities (ILAs), subclinical abnormalities detectable on HRCT, may help to diagnose ILD in an earlier phase. ILAs have been studied in rheumatoid arthritis [34]. One study in early SSc-associated ILD demonstrated ground glass opacities that later progressed to an NSIP pattern [35]. Although HRCT is an essential tool for the diagnosis of ILD, it exposes patients to radiation; a too close follow-up in asymptomatic patients can expose them to a high radiation risk, notably for younger patients. Low-dose computed tomography (CT) may be used for this purpose but should be validated in large cohorts. A crucial point is that patients with early ILD may have normal lung volumes, even those who show radiological abnormalities on HRCT. Two studies found that over 60% of patients with SSc and ILD who were diagnosed using HRCT had normal spirometry [36,37]. Therefore, pulmonary function tests cannot be used as a screening test for asymptomatic patients [38]. Since pulmonary function tests (PFT) are not sensitive enough in early SSc to be chosen as a screening procedure and HRCT exposes patients to radiation risk, a sensitive and risk-free procedure is needed to diagnose early ILD in SSc patients, so that prognostic evaluation can be made and appropriate therapies can be started in high-risk patients; ML may be the optimal solution in the future. To this extent, in order to prove its usefulness in this specific clinical domain, we used ML to predict early lung involvement in asymptomatic SSc patients. Some other works already applied this approach to the clinical question, with positive outcomes suggesting the possibility to enhance the application of such methodology in the specific field of SSc diagnosis. Clinical data were used by Jamian and colleagues [16], who were able to detect the presence of SSc in a large dataset derived from an electronic health record. On the other hand, biochemical markers, including skin biopsies, were employed in other works (e.g., [17]), further demonstrating AI usefulness in SSc diagnosis, but at the same time leaving room for some improvement in terms of the type of data collected and merged to train ML algorithms. Similar to our approach, a recent work found DLCO as an independent predictor for ILD, checked by lung ultrasound, at one year, even if their work was limited to building a multivariate regression model rather than trying different ML algorithms, which were, however, the scope of the present investigation [39].

With respect to the majority of published works, our data were extracted by a single dataset, thus avoiding possible “batch effects” due to different data sources for the training and test set. Obviously, as SSc is a rare disease, whose incidence accounted for 7.2–33.9 cases per 100,000 individuals in Europe [40], it is rather difficult to obtain larger datasets from a single center. On the other hand, the recruitment of patients from different clinical institutions would represent an additional bias similar to what has been stated above, therefore being undesirable at this stage. Therefore, we had to manage the trade-off between possible biases and the availability of a smaller dataset, which would have carried out further issues including likely overfitting and, lastly, low generalizability, which remained as the main limitations of our approach. Furthermore, clinical and instrumental data were employed here to this extent for the first time. Notably, to our results, total lung capacity was estimated to be the best element to predict lung involvement, especially when other spirometry parameters are also studied, including the FEV1 and FVC, as well as the diffusing capacity for carbon monoxide, the latter found to be predictive also by Pitsidianakis and colleagues [39]. As such, useful information can also be extracted from impedance pH monitoring, notably for the estimation of MNBI3, MNBI5 and MNBI7. Such results would enable the clinician to pick the exams with the highest predictive values, excluding ones with worse characteristics, saving time and money and reducing the burden or annoyance brought to the patient.

In terms of ML, the application of supervised algorithms was methodologically needed due to the presence of a known output driving the algorithm to learn the possible input-output associations. Overall, one of the main limitations of our study is represented by the small number of patients included in the dataset. Indeed, despite having applied a resampling method with random observations added to the dataset, it is essential to report that ML algorithms are more efficient when a high number of data are available for training and testing. Future works should then make profitable use of larger datasets for this purpose.

From the model’s point of view, taking into account their pros and cons, random forest was seen to ensure good performances in terms of prediction error, performing the regression task on a relatively low number of variables, making the overall model simple enough to be used by the clinician. However, its higher computational burden with respect to the other models make it poorly usable in cases where fast response times are requested, or where computational load might represent a significant constraint. In this regard, once established that the model owns optimal properties in terms of correct classification (or regression) with respect to other methods, the usage of R-based packages and libraries other than *caret*, including the *tidymodels* library, would allow the data scientist to reduce the complexity of the classifier, in turn saving time and computational resources in constrained use cases.

## 5. Conclusions

The present work was carried out investigating a small cohort of subjects with SSc and attempted to predict early lung involvement according to clinical and instrumental tools via the application of ML models. According to the results obtained, ML models can be used for predicting early lung involvement in SSc patients. Thanks to this approach, we found that spirometry parameters could predict ILD in our cohort. This was even more accurate if used along with impedance pH monitoring. Risk-free and sensitive screening methods are needed for early detection of lung involvement in SSc patients and ML could be the answer to this question in the future: indeed, ML could help to diagnose interstitial lung disease in SSc patients in early stages and could be a useful tool to identify the patients who may need early therapy, improving their quality of life and reducing hospitalizations or unnecessary exams, with a consequence of saving money for the national health systems and burden reduction for the patients. In addition, future studies can be performed applying ML to classify SSc with respect to other autoimmune disorders or medical conditions with somewhat similar clinical characteristics.

## Figures and Tables

**Figure 1 diagnostics-11-01880-f001:**
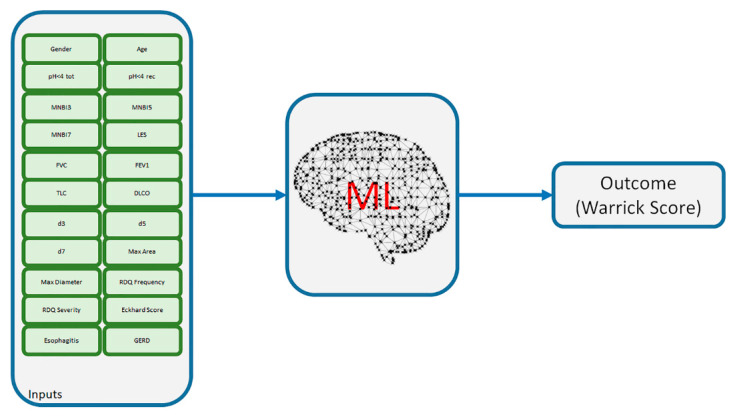
Overall view of the inputs and outcome of the ML algorithms.

**Figure 2 diagnostics-11-01880-f002:**
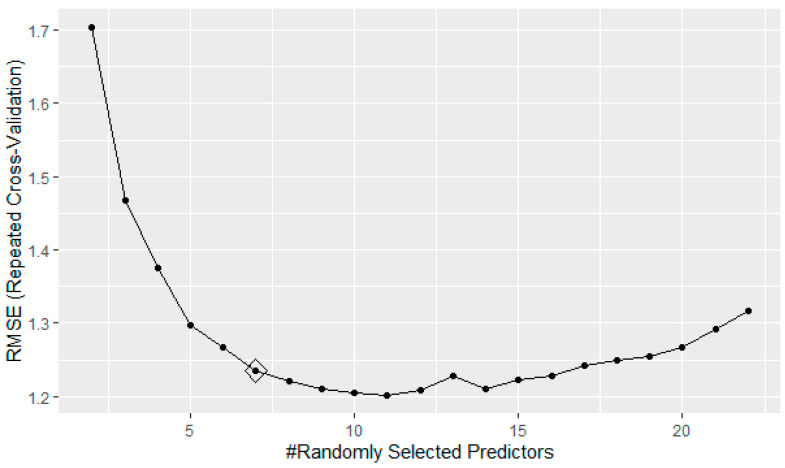
RMSE trend varying with the number of predictors used for the model.

**Table 1 diagnostics-11-01880-t001:** Dataset variables characteristics (DLCO: diffusing capacity for carbon monoxide; FEV1: forced expiratory volume in the 1st second; FVC: forced vital capacity; GERD: gastro-esophageal reflux disease; LES: lower esophageal sphincter; MNBI3: mean nocturnal basal impedance at 3 cm; MNBI5: mean nocturnal basal impedance at 5 cm; MNBI7: mean nocturnal basal impedance at 7 cm; RDQ: reflux disease questionnaire; TLC: total lung capacity).

Feature (u.m.)	Type	Mean ± SD	Range (Min-Max)	Normal Distribution
Gender	Dichotomous	N/A	N/A	N/A
Age (years)	Continuous	63.5 ± 11.7	35–89	Yes
pH < 4 tot (%)	Continuous	7.5 ± 10.0	0–38.9	No
pH < 4 rec (%)	Continuous	8.8 ± 16.2	0–72.1	No
MNBI3 (Ω)	Continuous	1166.7 ± 974.9	135.6–4345.9	No
MNBI5 (Ω)	Continuous	1337.1 ± 1180.6	133–4490.3	No
MNBI7 (Ω)	Continuous	1385.5 ± 1239.3	114.6–4507.7	No
LES (mmHg)	Continuous	15.4 ± 11.6	−3–+55	No
FVC (%)	Continuous	104.3 ± 30.3	0–158	Yes
FEV1 (%)	Continuous	96.7 ± 27.0	0–140	No
TLC (%)	Continuous	87.2 ± 29.3	0–151	No
DLCO (%)	Continuous	69.2 ± 23.1	0–134	Yes
d3 (mm)	Continuous	8.1 ± 8.3	0–30	No
d5 (mm)	Continuous	6.8 ± 7.5	0–25	No
d7 (mm)	Continuous	8.3 ± 8.4	1–30	No
Max area (mm^2^)	Continuous	122.2 ± 127.1	1.3–625	No
Max diameter (mm)	Continuous	14.5 ± 6.9	1–34	Yes
RDQ frequency (score)	Continuous	8.8 ± 8.2	0–30	No
RDQ severity (score)	Continuous	11.1 ± 9.3	0–28	No
Eckhard score (score)	Ordinal	0.97 ± 0.97	0–3	No
Esophagitis (Y/N)	Dichotomous	N/A	N/A	N/A
GERD (Y/N)	Dichotomous	N/A	N/A	N/A
Warrick score (score)	Continuous	9.3 ± 7.7	0–27	No

**Table 2 diagnostics-11-01880-t002:** Classifiers comparison in terms of RMSE with respect to the test set, for the selected model, deviating 1SE from the minimum error.

Classifier	Hyper-Parameter(s)	Hyper-Parameter(s) Value(s) Range	Hyper-Parameter(s) Optimal Value(s)	RMSE		R-Squared	
				Test Set	Training Set	Test Set	Training Set
LASSO	fraction	0–1	0.718	4.091	4.074	4.102	4.095
RIDGE	lambda	0–1	0.012	4.090	4.013	4.111	4.042
Elastic net	fraction, lambda	0–1	0.765 (fraction), 0 (lambda)	4.074	4.033	4.121	4.123
CART	cp	0–1	0.004	2.169	2.264	7.810	7.533
Random forest	mtry	1–22	7	0.810	0.425	0.619	0.485

**Table 3 diagnostics-11-01880-t003:** Classifiers comparison in terms of time elapsed, memory used, number of variables included in the optimal model.

Classifier	Time Elapsed (s)	Memory Used (MB)	Number of Variables
LASSO	34.09	0.303	20
RIDGE	655.06	11.5	2
Elastic net	108.14	11.6	20
CART	143.92	1.49	9
Random forest	1422.39	5.99	7

## Data Availability

Anonymized data available upon request.

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
