# Peer review of "A Machine Learning Application to Predict Early Lung Involvement in Scleroderma: A Feasibility Evaluation"

_diagnostics, 2021, doi:10.3390/diagnostics11101880_

Round 1
Reviewer 1 Report
1. The existing literature should be classified and systematically reviewed, instead of being independently introduced one-by-one.
2. The abstract is too general and not prepared objectively. It should briefly highlight the paper's novelty as what is the main problem, how has it been resolved and where the novelty lies?
3. The 'conclusions' are a key component of the paper. It should complement the 'abstract' and normally used by experts to value the paper's engineering content. In general, it should sum up the most important outcomes of the paper. It should simply provide critical facts and figures achieved in this paper for supporting the claims.
4. For better readability, the authors may expand the abbreviations at every first occurrence.
5. The author should provide only relevant information related to this paper and reserve more space for the proposed framework.
6. However, the author should compare the proposed algorithm with other recent works or provide a discussion. Otherwise, it's hard for the reader to identify the novelty and contribution of this work.
7. The descriptions given in this proposed scheme are not sufficient that this manuscript only adopted a variety of existing methods to complete the experiment where there are no strong hypothesis and methodical theoretical arguments. Therefore, the reviewer considers that this paper needs more works.
8. The related works section is very short and no benefits from it. I suggest increasing the number of studies and add a new discussion there to show the advantage. Following studies can be considered
- Lung nodules detection using semantic segmentation and classification with optimal features
Author Response
We thank the reviewer for their valuable comments. Please, find our point-to-point response, with reviewer's comments in plain text and our response in italics.
Reviewer 1
- The existing literature should be classified and systematically reviewed, instead of being independently introduced one-by-one.
Thank You. We logically restructured the Introduction highlighting the current state of methods employed for screening purposes related to SSc and associated hallmarks and providing hints about the drawbacks of the current methods, discussing how technological advances, including ML, can bring positive solutions to the clinicians.
- The abstract is too general and not prepared objectively. It should briefly highlight the paper's novelty as what is the main problem, how has it been resolved and where the novelty lies?
Thank You. We modified the Abstract accordingly.
- The 'conclusions' are a key component of the paper. It should complement the 'abstract' and normally used by experts to value the paper's engineering content. In general, it should sum up the most important outcomes of the paper. It should simply provide critical facts and figures achieved in this paper for supporting the claims.
Thank You. We revised accordingly.
- For better readability, the authors may expand the abbreviations at every first occurrence.
Thank You. Double-checked and corrected.
- The author should provide only relevant information related to this paper and reserve more space for the proposed framework.
Thank you. We agree with you and added more information about the proposed framework, and related literature, in the Abstract, Introduction and Conclusions.
- However, the author should compare the proposed algorithm with other recent works or provide a discussion. Otherwise, it's hard for the reader to identify the novelty and contribution of this work.
Thank You. In the paragraph “Machine Learning” of the “Method” section, a brief explanation of the choice for selected algorithms was provided. Of course, every choice has pros and cons but, compared to other ML models, like Conditional Random Forest or Bayesian techniques, our algorithms were, overall, simpler and featured a reduced computational load, making them the ideal choice for our investigation.
- The descriptions given in this proposed scheme are not sufficient that this manuscript only adopted a variety of existing methods to complete the experiment where there are no strong hypothesis and methodical theoretical arguments. Therefore, the reviewer considers that this paper needs more works.
Thank You. We tried to argument the reason why such methods have been used based on the sample size and the need for cost-effective methods from a computational point of view (see previous response).
- The related works section is very short and no benefits from it. I suggest increasing the number of studies and add a new discussion there to show the advantage. Following studies can be considered
Lung nodules detection using semantic segmentation and classification with optimal features
Thank You. We added some literature dealing with the specific topic of SSc in the Introduction section.
Reviewer 2 Report
I read with interest this manuscript on the use of machine learning for the prediction of early involvement of the lung in scleroderma. This is an interesting idea but specific considerations need to be taken into account t prior to publication.
- I don't understand why the outliers were removed without any reasoning. Please provide an explanation because this could be an important drawback of your work.
- How many artificial observations did you add to your training set?
- what is the exact number of artificial and real observations for each one of your training and test set?
- You need to provide ROC curves for the performance of your models.
- for the best performing method please provide a list of all the parameters used
- Figure 3 is redundant. It is not relevant to the conclusions of the paper and represents only technical information that is already included in the text. please remove.
- Please provide an AUC for each of the models and statistically compare the AUCs of individual models (e.g. with DeLong's method).
- Other manuscripts have recently used regression to correlate measurable clinical and imaging parameters to Warrick scores of ILD such as Pitsidianakis et al. doi: 10.1002/jum.15790. You need to compare your methodology to theirs in your discussion and the predictors they found to your predictors.
- "Furthermore, clinical and instrumental data were employed here for the first time to 327 this extent" this is not a paragraph. please amend.
- please provide references for the information you present on different models (m&m section).
- how did you optimise the hyperparameters of the model?
- R-studio is a software but is just a way to code in R which is a programming language. Please explain in your materials and methods.
- I would like to see how a tree-based method such as XGboost performs in your data. XGboost is less prone to overfitting than random forests. XGboost is considered the optimal algorithm for tabular data like yours and has won the most AI competitions among all other algorithms (apart from deep learning).
- The number of trees for RF should have been limited to the point that the model does not improve. Otherwise the more the trees the more the overfitting. please change.
- The text should be revised by a native speaker.
Author Response
We thank the reviewer for their valuable comments. Please, find our point-to-point response, with reviewer's comments in plain text and our response in italics.
Reviewer 2
I read with interest this manuscript on the use of machine learning for the prediction of early involvement of the lung in scleroderma. This is an interesting idea but specific considerations need to be taken into account t prior to publication.
I don't understand why the outliers were removed without any reasoning. Please provide an explanation because this could be an important drawback of your work.
Thank you, our mistake in the previous version. Now, we correctly specified our outlier treatment rationale in the Machine Learning sub-section of the Methods.
How many artificial observations did you add to your training set?
what is the exact number of artificial and real observations for each one of your training and test set?
Thank you. The number of observation was, overall 38 (real) + 190 (artificial). The training set was composed of 182 observations (30 + 152) and the test set of the remaining 46 (8 + 38).
You need to provide ROC curves for the performance of your models.
We did not use ROC curves since we performed a regression task, not classification, and our outcome was not represented by a binary variable.
for the best performing method please provide a list of all the parameters used
Thank You. The Random Forest method was the one providing the best performances according to the error minimization. As reported in the Results section, the parameters used by the model were: i) Total Lung Capacity (TLC), ii) Mean Nocturnal Basal Impedance at 3 cm (MNBI3), iii) Diffusing Capacity for Carbon Monoxide (DLCO), iv) Forced Expiratory Volume in the 1st second (FEV1), v) Forced vital capacity (FVC), vi) Mean Nocturnal Basal Impedance at 5 cm (MNBI5), vii) Mean Nocturnal Basal Impedance at 7 cm (MNBI7).
Figure 3 is redundant. It is not relevant to the conclusions of the paper and represents only technical information that is already included in the text. please remove.
Thank You. Done.
Please provide an AUC for each of the models and statistically compare the AUCs of individual models (e.g. with DeLong's method).
Thank You. As for the ROC, we did not use AUC because of the regression task being the target of our work, and given that the models used were trained for regression and not for classification purposes.
Other manuscripts have recently used regression to correlate measurable clinical and imaging parameters to Warrick scores of ILD such as Pitsidianakis et al. doi: 10.1002/jum.15790. You need to compare your methodology to theirs in your discussion and the predictors they found to your predictors.
Thank You. It was our mistake to not having taken into account this work. We added it in the Discussion section and compared their results to ours, as suggested. Thank you again.
"Furthermore, clinical and instrumental data were employed here for the first time to 327 this extent" this is not a paragraph. please amend.
Thank You. Done.
please provide references for the information you present on different models (m&m section).
Thank You. References included.
how did you optimise the hyperparameters of the model?
Thank You. The hyperparameters were set in a large range of values (basically all possible values for the LASSO, RIDGE, Elastic Net and CART, values ranging from 2 to n-1, where n were the overall number of input variables, for RF). Then, we picked the value of the hyperparameter corresponding to the minimum value of RMSE plus 1 Standard Error to reduce the likelihood of overfitting.
R-studio is a software but is just a way to code in R which is a programming language. Please explain in your materials and methods.
Thank You. Done.
I would like to see how a tree-based method such as XGboost performs in your data. XGboost is less prone to overfitting than random forests. XGboost is considered the optimal algorithm for tabular data like yours and has won the most AI competitions among all other algorithms (apart from deep learning).
Thank You. We are aware of the noticeable advantages of the XGboost algorithm. However, the main drawback of our dataset is represented by the extremely low sample size available. This is due to several factors, among which the clinical constraints represented by the COVID-19 pandemic, that have prevented the clinicians from being able to collect data from more patients. Being among the main drawbacks of the XGboost the fact that the number of training samples is quite limited, especially if compared with the large enough amount of features our dataset is composed of, that was the main reason why we decided not to implement this method in our dataset.
The number of trees for RF should have been limited to the point that the model does not improve. Otherwise the more the trees the more the overfitting. please change.
Thank You. You are totally right, but, as mentioned in the final part of the Results, in the present work we decided to use the R caret package, having the possibility to train different sorts of models and comparing them fairly. The package does not allow selecting the number of trees for RF, and sets the parameter to 500, that drives to possible overfitting. This was anyway presented as a limitation of the approach adopted (end of the Discussion section).
The text should be revised by a native speaker.
Thank You. Done.
Reviewer 3 Report
This paper proposed to utilize machine learning technique to analyze high resolution computed tomography, pulmonary function tests, esophageal pH impedance test, esophageal manometry and reflux disease questionnaires and predicted Warrick score for patients with Systemic sclerosis (SSc)
Dataset collection from 38 patients at the Department of Internal Medicine, University of Genoa, and at the Radiology Unit, IRCCS Policlinico San Martino, Genoa, is used for 10-fold cross-validation. That is, in each trial, 4 samples are for testing and the rest 34 are for training. Specifically for each sample, there are 22 input variables and the prediction output is a continuous Warrick score.
However, there are some concerns that shall be addressed to improve the quality of the work.
First, in the introduction, how this tool could benefit the current system shall be discussed in more detail.
Secondly, the motivation of using these parameters (in Table 1) to predict the score shall be explained in detail. (Not sure if there is any reference available.)
Also, the correlation between these parameters shall be investigated in order to analyze the utility of these parameters.
Lastly, the contribution of this work shall be clearly explained. Simply use the existing machine learning model to perform regression analysis on a dataset is a weak contribution to this journal.
Also, the authors mentioned that Pulmonary function tests are not sensitive enough to be used for screening purposes. so what is the level of sensitivity required by the task?
- Not sure what is 228 observations mentioned in line 177.
Author Response
We thank the reviewer for their valuable comments. Please, find our point-to-point response, with reviewer's comments in plain text and our response in italics.
Reviewer 3
This paper proposed to utilize machine learning technique to analyze high resolution computed tomography, pulmonary function tests, esophageal pH impedance test, esophageal manometry and reflux disease questionnaires and predicted Warrick score for patients with Systemic sclerosis (SSc)
Dataset collection from 38 patients at the Department of Internal Medicine, University of Genoa, and at the Radiology Unit, IRCCS Policlinico San Martino, Genoa, is used for 10-fold cross-validation. That is, in each trial, 4 samples are for testing and the rest 34 are for training. Specifically for each sample, there are 22 input variables and the prediction output is a continuous Warrick score.
However, there are some concerns that shall be addressed to improve the quality of the work.
First, in the introduction, how this tool could benefit the current system shall be discussed in more detail.
Thank You. We logically restructured the Introduction highlighting the current state of methods employed for screening purposes related to SSc and associated hallmarks and providing hints about the drawbacks of the current methods, discussing how technological advances, including ML, can bring positive solutions to the clinicians.
Furthermore, ML can confirm that the possibility exists for predicting ILD, anticipating functional signs shown by the spirometry and pH-impedentiometry. This would enlarge the indication of HRTC foreseen in the international guidelines (Kowal-Bielecka et al., Ann Rheum Dis. 2017; van den Hoogen et al., Ann Rheum Dis. 2013).
Secondly, the motivation of using these parameters (in Table 1) to predict the score shall be explained in detail. (Not sure if there is any reference available.)
Thank You. The parameters used for the analysis come from the exams foreseen by the guidelines for SSc management (see references cited in the previous response).
Also, the correlation between these parameters shall be investigated in order to analyze the utility of these parameters.
Thank you. Of course, many of these parameters were strongly correlated to each other, even simply because of their nature of being extracted by the same clinical examinations. On one side, the strong correlation between them is a significant methodological hurdle for the good outcome of the ML approach, therefore some of them were not used to train the ML models employed in the manuscript as they would create issues to the models themselves. On the other side, the study of all the parameters mentioned in the methodological part would enable the clinician undergoing a precise, exhaustive clinical characterization of the patient. Overall, the analysis conducted revealed the capability of the HRCT, even prior to spirometry, to predict the pulmonary damage, thus enabling early treatment of SSc-associated events.
Lastly, the contribution of this work shall be clearly explained. Simply use the existing machine learning model to perform regression analysis on a dataset is a weak contribution to this journal.
The application of various ML models for regression analysis (probably, the most famous ones in the ML domain applied to this specific investigation) in a dataset composed by individuals assessed via both clinical and instrumental tools within the SSc framework represents a novelty in terms of discovering how impactful are such analyses for prediction of lung involvement in this specific condition, possibly paving the way for a more precise use of clinical and instrumental tools for such purpose, with a significant money saving for national health systems in terms of avoidable, unnecessary exams.
Also, the authors mentioned that Pulmonary function tests are not sensitive enough to be used for screening purposes. so what is the level of sensitivity required by the task?
HRCT and spirometry are two kinds of exams with different indications and aims. Anatomical damage occurs earlier than the functional one, therefore the patient should be treated very early in order to limit, as much as possible, the onset of clinical symptomatology.
- Not sure what is 228 observations mentioned in line 177.
That is the number of observation derived from the number of the patients included in the dataset used for ML purposes, and resampled to try overcoming the major limitation concerning the low sample size for the current study.
Round 2
Reviewer 1 Report
The manuscript is fine tuned and can be accepted.
Reviewer 2 Report
the authors have sufficiently addressed all my concerns
Reviewer 3 Report
Thank you for addressing all the concerns.